# UVMS Trajectory Tracking Based on RBFNN and Sliding Mode Control

Huiyi Luo
*Fuzhou Institute of Oceanography, Fuzhou University,*
Fuzhou 350108, China
*College of Mechanical Engineering and Automation, Fuzhou University,*
Fuzhou 350108, China
18278811826@163.com

Weilin Luo
*Fuzhou Institute of Oceanography, Fuzhou University,*
Fuzhou 350108, China
*College of Mechanical Engineering and Automation, Fuzhou University,*
Fuzhou 350108, China;
wlluo@fzu.edu.cn

Yuanjing Wang
*College of Mechanical Engineering and Automation, Fuzhou University,*
Fuzhou 350108, China
xyjw325@163.com

*Abstract*—**This article spells out the UVMS trajectory tracking control problem under electric drive. Firstly, based a claim on Radial Basis Function Neural Networks (RBFNN) and Nonsingular Fast Terminal Sliding Mode (NFTSM) methods, the tracking strategy for UVMS is designed. Further, for singularity problem, a saturation-based tracking controller is obtained by means of the methods mentioned above. Lyapunov design is adopted to guarantee the asymptotic stability of the proposed controller. Simulation results show that the tracking consequence of NN-NFTSM is more thoroughly than PD approach and NN approach. The validity and advantages of the proposed controller is testified.**

*Keywords—UVMS, electric drive, trajectory tracking, fast nonsingular terminal sliding mode, RBF neural network*

## I. INTRODUCTION

Underwater Vehicle-Manipulator Systems (UVMS), which can control the underwater manipulator to complete the underwater task instead of human beings, is an effective means to develop underwater ocean energy at present. Usually, UVMS is constituted if there are n-link manipulators connected to an underwater robot for instance ROV (Remotely Operated Vehicle) and AUV (Autonomous Underwater Vehicle). As a vital tool of underwater vehicle, UVMS is pretty significant for these underwater operations, for example underwater real time shooting, underwater target reconnaissance and surveillance, marine resource exploitation, marine bioprospecting, etc. UVMS plays a supporting role in various marine underwater missions, and becomes the research focus of many scholars.

How to settle the uncertainties in an underwater condition, like current, oceanic internal wave, is the biggest challenge for an UVMS to reach an ideal performance controller. For this reason, the effectiveness and robustness of controller is pretty crucial. Xu, et al. adopted fuzzy based control techniques to study a 6-DOF AUV which has a 3-DOF on-board manipulator [1]. Wei, et al. have applied a nonlinear disturbance observation for an UVMS to evaluate the external unpredictable disturbance in real time, and an adaptive sliding mode approach is utilized for compensating things [2]. Mobayen et al. adopted a continuous nonsingular fast terminal sliding mode control with timing delay evaluation, which can make full sure the satisfactory of tracking control performance and the sufficiency of robustness on an UVMS [3]. Wang et al. have selected the control plan which mixed sliding mode control and adaptive fuzzy control to constitute a multi-strategy fusion control that addressed the motion variable control issue of UVMS [4]. Luo et al. applied neural networks to a 3-link UVMS's tracking, the robustness of controller is verified by compared with PD control method [5]. Mofid et al. applied a fuzzy terminal sliding mode control approach with timing delay evaluation, which puts the focus on using fuzzy rules to adaptively fit the terminal sliding mode surface to eliminate the unpredictable internal and external disturbance running on manipulator [6]. Woolfrey et al. applied model predictive control plan to study kinematics things on UVMS which is affected by fluctuations, and the results show that the approach has excellent predictive consequence [7]. Han and Chung exposed an approach which uses restoring moments to explore the motion control under external disturbance of an UVMS [8].

This article proposes a fast nonsingular terminal sliding mode cascade controller combined with RBF neural network method for manipulator control problem of UVMS. Due to the interaction induced by vehicle and manipulator, there is an external disturbance working on UVMS, which is the main source of external disturbance. Lyapunov approach is applied to verify the stability of the cascade controller. The effectiveness and robustness of the controller designed in this article is guaranteed by numerical simulation.

## II. PROBLEM FORMULATION

When UVMS moves to working area, it is sometimes necessary for the underwater robot body to maintain a stable hover state while the mechanical arm works according to the work requirements. At this time, the body-fixed reference coordinate system attached to the underwater vehicle body can be viewed as the inertial reference coordinate, which is constructed with the earth, and the motion of the entire UVMS can be regarded as the motion control of underwater robotic manipulators considering disturbance.

Since the influence of underwater robot on underwater robotic manipulators is difficult to be expressed by mathematical model, it can be regarded as disturbance on underwater robotic manipulators. The nonlinear dynamics of the underwater robotic manipulators is written as

$$M(q)\ddot{q} + C(q,\dot{q})\dot{q} + D(q,\dot{q})\dot{q} + G(q) + \Delta = \tau_{ms} \quad (1)$$

---

Corresponding Author: W. Luo
This work was supported by the Natural Science Foundation of Fujian Province, China through Grant 2023J011572, and Fuzhou Institute of Oceanography through Grants 2021F11 & 2022F13.

where $\Delta$ denotes the uncertainty induced by the interaction of underwater vehicle and manipulator, $M$ denotes the inertial matrix, $C$ denotes the Coriolis-centripetal matrix, $D$ denotes the water resistance coefficient matrix, $G$ denotes the equivalent gravity vector matrix, $\tau_{ms}$ denotes the input of control.

Fig. 1 displays underwater robotic manipulator combined with underwater vehicle to form a three-link UVMS. Fig. 1 shows the starting position of the underwater robotic manipulator, in which the joint at the hinge of the connecting rod is driven by a motor, so as to achieve the operational requirements of the three degree of freedom underwater robotic manipulator.

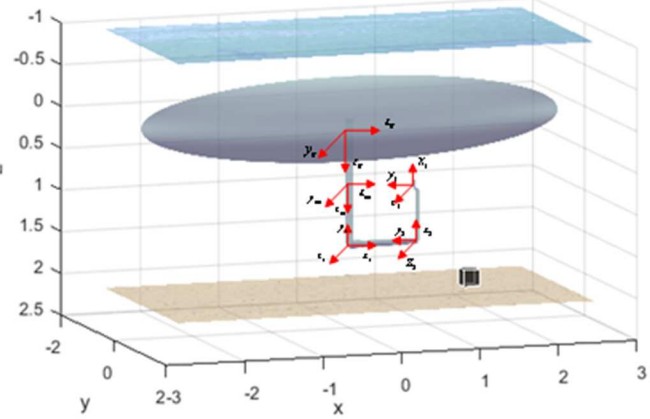

Fig. 1 Three-link manipulator UVMS

Since the underwater robotic manipulator's joint is run by a DC motor, the motor driving force can be described as

$$\tau_{me} = K_{me}I \qquad (2)$$

where $I$ denotes the electrical current, $K_{me}$ denotes the coefficient matrix during the process of electrical current change to torque.

The dynamics of the electrical circuit can be described as

$$\tau_e = L_e \dot{I} + R_e I + K_e \dot{q} \qquad (3)$$

where $\tau_e, L_e, R_e$ denotes the vector matrix of the motor coil's voltage, inductance and resistance, respectively. $K_e$ denotes the constant matrix of its voltage.

Then, a cascaded system containing the subsystem of machinery and electricity consists of Equations (1) and (3).

## III. CONTROLLER DESIGN

### A. NN Based Controller

According to Equation (2), an ideal trajectory design is carried out for the desired joint angle of the underwater robotic manipulator. By defining the desired joint angle as $q_d$ and considering Equations (1) and (3), the desired input signal of the electrical current can be described as

$$I_d = K_{me}^{-1} \left( M\ddot{q}_d + C\dot{q} + D\dot{q} + G + \Delta + \tau_1 \right) \qquad (4)$$

where $\tau_1$ denotes the auxiliary controller for dynamics of underwater robotic manipulator. Similarly, the auxiliary controller of electrical system can be designed as

$$\tau_e = R_e I_d + K_e \dot{q}_d + \tau_2 \qquad (5)$$

where $\tau_2$ represents the auxiliary controller for electrical system.

Further, define joint tracking error as

$$e = q_d - q \qquad (6)$$

To guarantee convergent quality, design the fast terminal sliding surface as

$$s = \dot{e} + \alpha_1 sign^{\gamma_1}(e) + \alpha_2 sign^{\gamma_2}(e) \qquad (7)$$

where $sign^{\Delta}(\cdot) = |\cdot|^{\Delta} sign(\cdot)$, $\gamma_1 \geq 1$, $0 \leq \gamma_2 \leq 1$, $\alpha_1$ and $\alpha_2$ are introduced as positive gain matrix.

Derivative of fast terminal sliding surface is

$$\dot{s} = \ddot{e} + (\alpha_1 \gamma_1 |e|^{\gamma_1 - 1} + \alpha_2 \gamma_2 |e|^{\gamma_2 - 1})\dot{e} \qquad (8)$$

To facilitate calculation, auxiliary variables are introduced as

$$\begin{cases} \vartheta = \alpha_1 sign^{\gamma_1}(e) + \alpha_2 sign^{\gamma_2}(e) \\ \mu = \alpha_1 \gamma_1 |e|^{\gamma_1 - 1} + \alpha_2 \gamma_2 |e|^{\gamma_2 - 1} \end{cases} \qquad (9)$$

Substituting Equation (9) into Equations (7) and (8) yields

$$\begin{cases} s = \dot{e} + \vartheta \\ \dot{s} = \ddot{e} + \mu\dot{e} \end{cases} \qquad (10)$$

Define electrical current error as $\eta = I_d - I$, one has

$$\begin{aligned} M(q)\dot{s} &= M\mu\dot{e} + M\ddot{e} \\ &= M\mu\dot{e} + M(\ddot{q}_d - \ddot{q}) \\ &= M\mu\dot{e} + K_{me}\eta - C\dot{e} + \Delta - \tau_1 \end{aligned} \qquad (11)$$

and

$$L\dot{\eta} = L\dot{I}_d - L\dot{I} = -R\eta - K(s - \vartheta) - \tau_2 + L\dot{I}_d. \qquad (12)$$

To reach the goal of letting the error Equation (11) and (12) attain to zero, Lyapunov design theorem is utilized and a positively definite Lyapunov function can be written as

$$V_1 = \frac{1}{2}(e^T e + s^T M s + \eta^T L \eta) \qquad (13)$$

The time derivative of keeps

$$\begin{aligned} \dot{V}_1 &= s^T(e + M\mu\dot{e} + C\vartheta + \Delta - \tau_1) - e^T\vartheta \\ &+ \eta^T[-R_e\eta + K_{me}s + K(s - \vartheta) + L_e\dot{I}_d - \tau_2] \end{aligned} \qquad (14)$$

Since Equation (14) contains nonlinear terms, and for the trajectory tracking control of underwater robotic manipulator, the nonlinear terms have an impact on the control results. For this reason, RBF neural network is adopted to estimate the nonlinear term. In detail, let

$$\begin{cases} f_1 = e + \mu M\dot{e} + C\vartheta + \Delta = W_1^T h_1(x) + \varepsilon_1 \\ f_2 = K_{me}s - R_e\eta + K_e(s - \vartheta) + L_e\dot{I}_d = W_2^T h_2(x) + \varepsilon_2 \end{cases} \qquad (15)$$

where $W_i, h_i, \varepsilon_i$ denote weights, inputs and regression errors, respectively.

The controllers $\tau_1$ and $\tau_2$ can be given as

$$\begin{cases} \tau_1 = W_{1e}^T h_1(x) + \alpha_1 M s \\ \tau_2 = W_{2e}^T h_2(x) + \alpha_2 L \eta \end{cases} \qquad (16)$$

where $W_{ie}$ denote updated weight matrices.

In order to achieve the excellent robustness of neural network controller, the weight is written as

$$\begin{cases} \dot{W}_{1e} = k_1 h_1(X_1)s^{\mathrm{T}} - k_2 W_{1e} \\ \dot{W}_{2e} = k_1 h_2(X_2)\eta^{\mathrm{T}} - k_2 W_{2e} \end{cases} \quad (17)$$

As pointed out [9], in a conventional sliding approach, since the item $\alpha_2 \gamma_2 |e|^{\gamma_2-1} \dot{e}$ in Equation (8) exists, it happens that $e_x \rightarrow 0$. In order to deal with the singular phenomena, one might use the following saturation

$$sat(v_z) = \begin{cases} v_z & |v_z| \leq \overline{w} \\ \overline{w}\,sign(v_z) & |v_z| \geq \overline{w} \end{cases} \quad (18)$$

where $v_z = \alpha_2 \gamma_2 |e|^{\gamma_2-1} \dot{e}$, $\overline{w}$ is a positive number.

Substituting Equation (18) into Equations (7), and replacing the fast terminal sliding surface (FTSM) to the nonsingular fast terminal sliding surface (NFTSM) yields

$$\dot{s}_2 = \ddot{e} + \alpha_1 \gamma \dot{e}_1 |e|^{\gamma_1-1} + v_z \quad (19)$$

Similarly, we can get

$$\begin{aligned} M(q)\dot{s}_2 &= M\alpha_1\gamma_1\dot{e}|e|^{\gamma_1-1} + v_z + M\ddot{e} \\ &= M\alpha_1\gamma_1\dot{e}|e|^{\gamma_1-1} + v_z + M(\ddot{q}_d - \ddot{q}) \\ &= M\alpha_1\gamma_1\dot{e}|e|^{\gamma_1-1} + v_z + K_{me}\eta - C\dot{e} + \Delta - \tau_1 \end{aligned} \quad (20)$$

To guarantee the stability, Lyapunov function is defined as

$$V_2 = \frac{1}{2}(e^T e + s_2^T M s_2 + \eta^T L\eta) \quad (21)$$

Its derivative is

$$\begin{aligned} \dot{V}_2 &= s_2^T(e + M\alpha_1\gamma_1 |e|^{\gamma_1-1}\dot{e} + v_z + C\vartheta + \Delta - \tau_1) - e^T\vartheta \\ &\quad + \eta^T[-R\eta + K_m s + k(s-\vartheta) + L\dot{I}_d - \tau_2] \end{aligned} \quad (22)$$

Combined with (15), the nonlinear term in the above expression can be cast as

$$f_3 = e + M\alpha_1\gamma_1 |e|^{\gamma_1-1}\dot{e} + C\vartheta + \Delta = W_{1N}^{\mathrm{T}}h_1(x) + \varepsilon_1 \quad (23)$$

The auxiliary controllers $\overline{\tau}_1$ and $\tau_2$ can be described as

$$\begin{cases} \overline{\tau}_1 = W_{1Ne}^T h_1(x) + \alpha_1 M s_2 - M v_z \\ \tau_2 = W_{2e}^{\mathrm{T}} h_2(x) + \alpha_2 L\eta \end{cases} \quad (24)$$

## B. Stability analysis

A Lyapunov function is designed as

$$V_3 = V_2 + \frac{1}{2k_1}\sum_{i=1}^{2}\left\|\tilde{W}_i\right\|_F^2 \quad (25)$$

where $\tilde{W}_i = W_i - W_{ie}$ represents weight error.

Its derivative is

$$\begin{aligned} \dot{V}_3 &\leq -2\alpha_0 V_3 + s^T\varepsilon_1 + \eta^T\varepsilon_2 - a(\alpha_1 s^T M s + \alpha_2 \eta^T L\eta) \\ &\quad + k_2\left(\sum_{i=1}^{2}(\tilde{W}_i, W_i)_F - a\sum_{i=1}^{2}\left\|\tilde{W}_i\right\|_F^2\right) \end{aligned} \quad (26)$$

in which $0 \leq a \leq 1$, $\alpha_0 = \min\{(1-a)\alpha_1,(1-a)\alpha_2,(1-a)k_2\}$.

In accordance with [10], it holds that

$$\dot{V}_2 \leq -2\alpha_0 V_2 + \lambda,(\lambda > 0) \quad (27)$$

Further, shrink Equation (27) as

$$\dot{V}_2 \leq -2\alpha_0 V_2 \leq 0 \quad (28)$$

From Equation (27) and (28), a conclusion can be made that the tracking system is stable. Thus, the effectiveness of controller in the control of UVMS underwater robotic manipulator is verified.

## IV. SIMULATION

For the purpose of testifying the validity and advantages of the designed tracking controller, i.e., NN based the nonsingular fast terminal sliding mode (NN-NFTSM) controller, comparison is conducted with traditional PD control and neural network control approaches. TABLE I. displays the parameters of robotic manipulator and controller.

TABLE I.    PARAMETERS OF THE UVMS

| Items | Rod1 | Rod2 | Rod3 |
|---|---|---|---|
| Length(m) | 1 | 1 | 1 |
| Mass(kg) | 1 | 1 | 2 |
| $L_e$ | 0.1 | 0.1 | 0.1 |
| $R_e$ | 1 | 1 | 1 |
| $K_e$ | 0.5 | 0.5 | 0.5 |
| $K_{me}$ | 1 | 1 | 1 |
| $\overline{w}$ | 0.5 | $\alpha_1,\alpha_2$ | 200 |
| $k_p,k_d$ | 300 | $k_1,k_2$ | 50,0.8 |

Since the underwater robotic manipulator is mounted on underwater vehicle to form the UVMS system, the first hinge of the underwater robotic manipulator has direct interference with the vehicle. In the simulation process, it is assumed that this interference is a transient interference signal: a force of 200N is applied to the vehicle at t =1.7s.

Fig. 2 displays the spatial tracking effect of UVMS end effector. It can be seen that the proposed NN-NFTSM controller is obviously better than traditional PD control and neural network control methods.

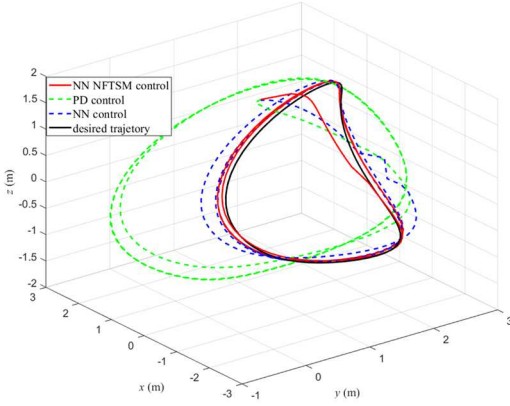

Fig. 2 Spatial tracking effect of UVMS end effector

Fig. 3 shows the results of joint angle tracking control. It is obvious to get the result that both the nonsingular fast terminal sliding mode surface based on neural network and the proposed sliding mode controller based on neural network have higher tracking stability than PD control.

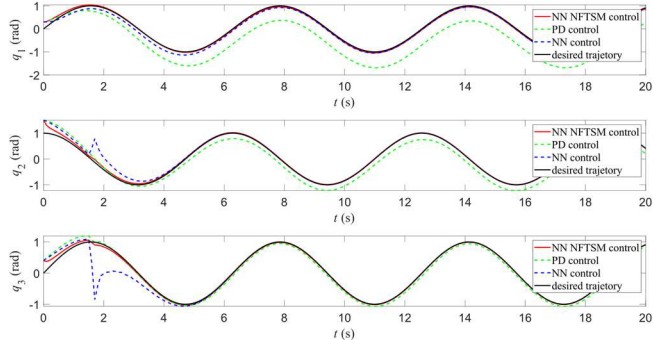

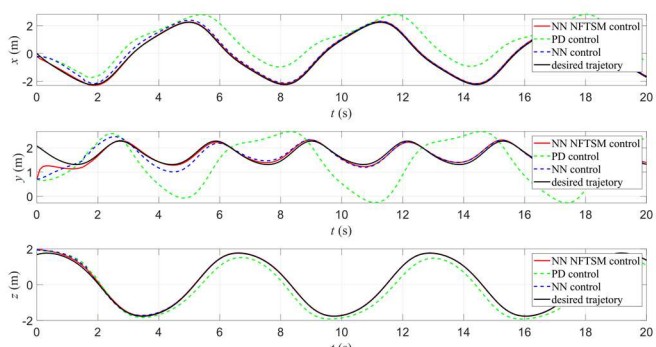

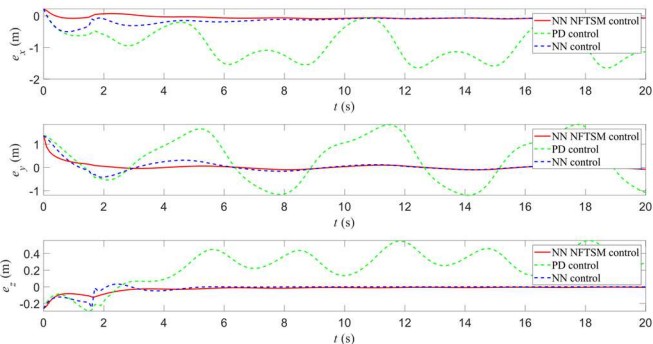

Fig. 3  Results of joint angle tracking control

In Fig. 4 and Fig. 5, the tracking effect of UVMS end effector in *x*, *y*, *z* directions are displayed. It is easy to get that in the method with neural network control, the tracking effect of three directions can reach stability. The proposed non-singular fast terminal sliding mode control method combined with the RBF neural network can track the desired trajectory more quickly and stably.

Fig. 4  Tracking effect of UVMS end effector in x, y, z directions.

Fig. 5  UVMS end effector tracking error

Fig. 6 ~ Fig. 8 show the comparison of MAE and RMSE under three control schemes. It is easy to get that NN-NFTSM has higher accuracy than RBF neural network (NN) and PD control.

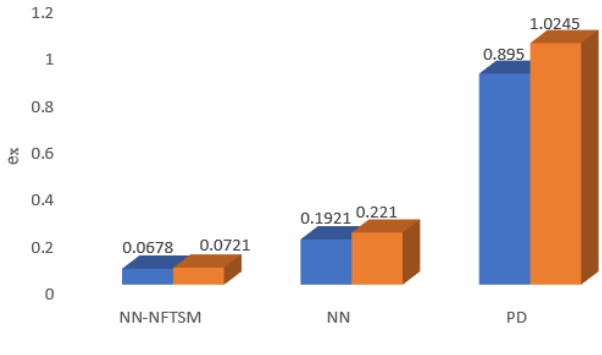

Fig. 6  Error in x direction

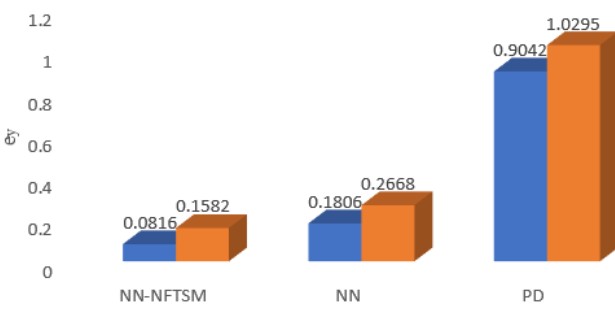

Fig. 7  Error in y direction

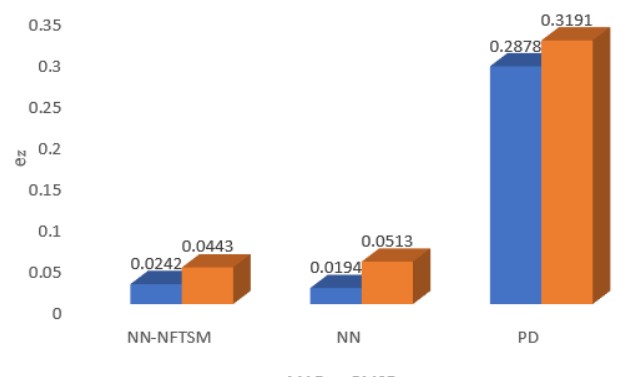

Fig. 8  Error in z direction

## V. CONCLUSION

In this article, a RBFNN based fast nonsingular terminal sliding mode controller is designed for UVMS. Singular items of the UVMS system are approximated by RBF neural network. Lyapunov design is selected to test the stability and feasibility of the proposed controller. It is proved that the convergence of tracking errors falls into a small zero neighborhood within finite time. Finally, the simulation results confirm that the proposed controller performs an excellent role in UVMS system.

## ACKNOWLEDGMENT

The work in the paper is partly supported by the Natural Science Foundation of Fujian Province of China, Grant 2023J011572, and partly supported by Fuzhou Institute of Oceanography, Grants 2021F11 & 2022F13.

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
