# OpenReview forum: "UVMS Trajectory Tracking Based on RBFNN and Sliding Mode Control"
_IEEE.org/ICIST/2024/Conference — IEEE ICIST 2024 Conference Submission_

### Official Review · Reviewer_rJoa · 2024-08-24
**In this article, a non-singular fast terminal sliding mode control, based on a Radial Basis Function (RBF) neural network, is designed for Underwater Vehicle-Manipulator Systems (UVMS). However, the following issues are noted:**

**Rating:** 7
**Confidence:** 4

**Review:**

(1) The images in the article are not clear and there are formatting issues with the captions.
(2) The presentation there are many grammar errors.
(3) Literature review is not enough.

---

### Official Review · Reviewer_57Wa · 2024-08-25
**minor repair**

**Rating:** 8
**Confidence:** 3

**Review:**

1. If equation (1) is not first proposed by the author, corresponding references should be added.
2. What does "q" mean？
3. For the convenience of readers' understanding, the author should provide more details in the stability analysis section.

---

### Official Review · Reviewer_UGjq · 2024-09-02
**The revised paper can be accepted after careful modifications.**

**Rating:** 6
**Confidence:** 5

**Review:**

This paper addresses the UVMS trajectory tracking control problem under electric drive.
The reviewer's comments are as follows:
1.The English grammar and format of this manuscript could be further polished and checked carefully.
2.The format of references is not uniform.
3.To enhance the quality of the manuscript, it is recommened refining the language to improve readability and ensuring that the concepts are communicated accurately.
4.The effectiveness of the method can be more clearly reflected by properly describing the simulation results.
5.Figures 2 to 5 are unclear; please revise and replace them.

---

### Decision · Program_Chairs · 2024-09-06

Accept (Oral)